# Design of a High Sensitivity Microwave Sensor for Liquid Dielectric Constant Measurement

**DOI:** 10.3390/s20195598

**Published:** 2020-09-29

**Authors:** Honggang Hao, Dexu Wang, Zhu Wang, Bo Yin, Wei Ruan

**Affiliations:** College of Electronic Engineering, Chongqing University of Posts and Telecommunications, Chongqing 400065, China; S180401004@stu.cqupt.edu.cn (D.W.); S180431033@stu.cqupt.edu.cn (Z.W.); yinbo@cqupt.edu.cn (B.Y.); ruanwei@cqupt.edu.cn (W.R.)

**Keywords:** dielectric constant, liquid, sensor, SRR, CSRR

## Abstract

In order to improve the sensitivity of liquid dielectric constant measurements, a liquid dielectric constant sensor based on a cubic container structure is proposed for the first time. The cubic container, which consists of a dielectric substrate with a split resonant ring (SRR) and microstrip lines, can enhance the electric field intensity in the measuring area. High sensitivity can be obtained from measuring the dielectric constant with the characteristics of the structure resonate. The research results show that the resonant frequency of the sensor is shifted from 7.69 GHz to 5.70 GHz, with about a 2 GHz frequency offset, when the dielectric constant of the sample varied from 1 to 10. A resonance frequency offset of 200 MHz for the per unit dielectric constant is achieved, which is excellent regarding performance. The permittivity of oil with a different metal content is measured by using the relation between the fitted permittivity and the resonant frequency. The relative error is less than 1.5% and the sensitivity of measuring is up to 3.45%.

## 1. Introduction

The accurate measurement of dielectric constants of solid and liquid materials has an important significance in medical, industrial, chemical and biological fields [1,2,3]. At present, the instruments used to measure the dielectric constant of objects are relatively expensive. The resonant cavity method [4,5], transmission reflection method [6,7,8] and free space method [9] are used to measure the relative dielectric constant. The resonant cavity method has the highest precision, but it has higher requirements for sample size and a narrower measurement range [10]. The transmission reflection method is easy to operate, and the measurement frequency range is larger, while the error is higher [11]. The free space method is mainly used to measure the dielectric constant of the millimeter-wave frequency band, but the area of the object to be measured needs to be large enough to ensure the measurement accuracy [12]. The measurement of permittivity by microwave method is converting the variation of the permittivity to the variation of the electromagnetic parameter of the sensor [13,14,15]. In recent years, the microwave method is getting more and more attention because of its advantages, including contactless detection, automatic detection, a wide application range of the detection objects, a fast detection rate and a long continuous working time [16,17]. At present, there are two major ways to measure the permittivity of a liquid solution using the microwave method, the submersible type and container type. The submersible type puts the microwave resonance device into the liquid under test (LUT) for measurement by treating the microwave resonance device as a probe [18,19]. For the container type, it adds the LUT in a container, and places the container in the place where the electric field resonates most strongly. Usually, the material of the container is quartz or a plastic tube. The LC resonance circuit on the metamaterial structure is loaded in Reference [20]. The LUT is placed on the plastic tube of the dielectric substrate to measure the dielectric constant. The measured frequency range was 2–3 GHz. Reference [21] loads the photonic band gap and variable capacitance on the substrate-integrated waveguide. The LUT in the plastic container is located in the middle of the sensor. The relative dielectric constant is calculated by using the cavity perturbation technology.

Because the resonance unit of the sensor is mostly planar, the electric field distribution in the container area is generally a two-dimensional section. The container type cannot fully make contact with the LUT as does the submersible type. It can effectively avoid the mutual contamination between the LUT and the sensor, but the electric field generated by the sensor could attenuate when it passes through the container. This attenuation leads to a reduction in the intensity of the electric field in the container and a significant decrease in the sensitivity of the sensor.

In order to improve the sensitivity of the microwave method to measure the dielectric constant of the LUT, a liquid dielectric constant microwave sensor based on the structure of the cube containers is proposed in this paper. The function of the complementary split resonant ring (CSRR), split resonant ring (SRR) structure and the microstrip line in the design is discussed and the electric field in the testing area is studied. Using the prepared sample, the accuracy and sensitivity of the proposed structure are investigated. The results show that the proposed microwave sensor has a high accuracy and sensitivity due to the design of the cubic structure.

## 2. Theoretical Analysis

According to the electromagnetic theory, the complex dielectric constant of an object can be divided into real and imaginary components [22]:(1)$$\begin{array}{c} \varepsilon =\varepsilon_{0}\varepsilon_{r}=\varepsilon_{0}(\varepsilon^{\prime }-j\varepsilon^{\prime \prime })\\ \tan\delta =\varepsilon^{\prime \prime }/\varepsilon^{\prime } \end{array}$$
where *ε*_0_ represents the dielectric constant of a vacuum; *ε_r_* is the relative dielectric constant; and tan*δ* is the tangent of the dielectric loss angle. The resonant frequency and electric field distribution of the sensor are fixed in a certain resonant mode. When the LUT is placed in the concentrated part of the electric field of the sensor, the electric field distribution of the sensor will be disturbed by the liquid to change its resonant frequency [23,24,25]. According to the change in resonance frequency, the relative dielectric constant of the LUT can be calculated by the formula. The calculation formula of the real part of the relative dielectric constant is as follows [26]:(2)$$\varepsilon^{\prime }=1+\frac{V_{0}}{V_{1}}\frac{\left(f_{0}-f_{1}\right)}{2f_{0}},$$
where *ε**ʹ* is the real part of the relative dielectric constant; *V*_0_ and *V*_1_ are the volume of the sensor and the LUT; and *f*_0_ and *f*_1_ represent the resonant frequency of the empty cavity and the resonant frequency of the filled cavity. When the physical dimensions of the sensor and the LUT are determined, $\frac{V_{0}}{V_{1}}$ is constant, then Equation (2) can be simplified into
(3)$$\varepsilon^{\prime }=1+A\frac{\Delta f}{2},$$
where
(4)$$\Delta f=\frac{\left(f_{0}-f_{1}\right)}{f_{0}}.$$

*A* is a constant, representing the effective volume ratio of perturbation. The larger *A* is, the more sensitive the sensor is.

## 3. Sensor Design

The key to ameliorate the performance of the dielectric constant measurement based on the microwave method is to optimize the distribution of the electric field in the measuring area. It can increase the effective volume ratio of the perturbation to load the SRR of the side wall of the container. Based on this, a liquid dielectric constant sensor based on a cubic container structure is proposed in this paper, as shown in Figure 1.

The container of the LUT is composed of the dielectric substrate loaded with the microstrip line and SRR structure. As the substrates of the container are relatively thin, the substrates of the different planes are connected through the welding points of the microstrip line. The capacity of the container is about 1.86 mL. To evade the contamination between the LUT and sensor, a replaceable plastic film with a thickness of 0.008 mm was added along the internal face of the container; a new one is applied per measurement. The influence of thin films has been considered in the simulation process to eliminate measurement errors. By quantitatively obtaining the liquid, the repeatability of the measurements is satisfied. The outer walls of the container are microstrip lines. The inner sides are four SRRs with the same structure as the bottom CSRR. The container transfers the electric field in the plane to the four sides through the coupling line.

The material of the sensor’s dielectric substrate and three-dimensional container is Rogers5880, with a relative dielectric constant of 2.2 and dielectric loss of 0.0009. The thickness of the dielectric substrate is 0.787mm. The thickness of the four side walls of the container is 0.127mm. The input and output ports are 50-ohm microstrip lines, and the middle part is a radiation patch loaded with the CSRR structure. The radiation patch and microstrip line are matched by the λ/4 step impedance (SIR).

The electromagnetic properties of the SRRs make them very easy to be coupled by the electric field or the magnetic field in the near field range. The SRR is composed of two concentric metal rings of different sizes inside and outside, with a gap at each symmetrical position in the center of both rings. Its structure and equivalent circuit are shown in Figure 2 [27].

The two metal rings of the SRR can be considered as equivalent inductance *L_SRR_*. The gap between the inner and outer rings can be regarded as equivalent capacitance *C_SRR_*. *C_SRR_* can be thought to be the series of capacitors between two half rings [28,29]. M represents the magnetic coupling between SRR and the microstrip line, and *Cs* is the capacitance between the SRR and microstrip line. The resonance frequency of the SRR can be obtained as follows:(5)$$f_{SRR}=\frac{1}{2\pi \sqrt{L_{SRR}(C_{SRR}+C_{S})}}$$

The equivalent capacitance and inductance are related to the size parameters of the resonance ring. The resonance frequency can be tuned by adjusting the size parameters of the SRR.

As a complementary structure of SRR, CSRR is also widely used in the design of microwave resonance devices. Its structure and equivalent circuit are shown in Figure 3. The symbols *L_CSRR_* and *C_CSRR_* represent the inductance and capacitance of the CSRR, respectively, and *L*_1_ is the line inductance; *C_S_* is the coupling capacitance between the CSRR and microstrip line. The orange part is the metal layer. White is the gap etched in the shape of the SRR. Under perfect circumstances, CSRR and SRR share the same parameters as well as resonate frequency. Thus, the resonate frequency of the CSRR can be tuned via its parameters [30].

The equivalent LC circuit model of the container-based sensor can be seen in Figure 4, where M_1_ and *C*_1_ represent the magnetic coupling and electric coupling between the parallel coupling lines.

According to the parameter extraction method reported in [31], the circuit parameters of the equivalent circuit model of Figure 4 are obtained. The comparison of the electromagnetic simulation and the equivalent circuit simulation is depicted in Figure 5.

The 3dB method to estimate the quality factor *Q* of the resonant cavity is based on the following equation [32,33]:(6)$$Q=\frac{f_{0}}{\Delta f_{3\mathrm{dB}}}$$
where *f*_0_ represents the resonance frequency and Δ*f*_3dB_ is the 3dB bandwidth of S_21_. According to Equation (6), the *Q*-factor of sensor is 73.33.

Figure 6 shows the electric field distribution of the sensor without the LUT at 7.689 GHz, which is the resonant frequency of the sensor when unloaded.

As can be seen from Figure 6, the electric field of the sensor is mainly concentrated at the CSRR in the middle of the dielectric substrate. In addition, due to the SRRs, the electric field gets stronger at the inner side of the container. In order to discuss the influence of microstrip lines and SRR on the electric field focusing effect on the container wall, the electric field distribution diagram of the sample area under different container structures is studied, as shown in Figure 7.

When there is no microstrip structure on the side wall of the container, the maximum electric field intensity in the liquid area to be measured is 2585 V/m, as shown in Figure 6. The highest electric field intensities are 6455 V/m and 4925 V/m, when only the microstrip lines or SRR are loaded. When the microstrip line and SRR were simultaneously loaded, the electric field intensity reached a maximum of 25,628 V/m. The simulation results illustrate that the electric field intensity can be significantly enhanced by loading the microstrip line and SRR resonance structure on the dielectric substrate of the container. The sensitivity of the sensor is improved in this way. When the LUT dielectric constant changes, the resonance frequency of the sensor will be shifted, as shown in Figure 8.

The simulation results show that the resonant frequency of the sensor is shifted from 7.689 GHz to 5.699 GHz, with a 1.99 GHz frequency offset, when the dielectric constant of the LUT varied from 1 to 10. In order to further study the influence on the dielectric constant sensitivity from the cubic structure, all the side walls were removed. The LUT is placed in glass or plastic containers on the CSRR structure. Considering the manufacture, the thickness of the two containers were set to 0.1 mm. The variation in the dielectric constant of the LUT and the resonant frequency are as shown in Figure 9.

As can be seen in Figure 9, the resonance frequency of the sensor corresponding to the glass container is shifted from 6.704 GHz to 5.916 GHz, with an offset of 788 MHz, and the PVC container is shifted from 6.784 GHz to 6.190 GHz, with an offset of 594MHz, when the dielectric constant of the LUT changes from 1 to 10. In addition, when the permittivity is larger than 6, the frequency offset caused by the change of permittivity is faint.

Comparing Figure 8 and Figure 9, the resonance frequency offset of the cubic container structure is much larger than that the ordinary container, which verifies that the three-dimensional electric field is better in sensing the dielectric property change of the LUT. Figure 10 illustrates that the resonant frequency curve varies with the permittivity. The red high light part represents the location of the resonant frequency point, and the white dotted line describes the curve of the resonance frequency change with a dielectric constant. The fitting relationship between the dielectric constant and resonance frequency (GHz) is listed as Equation (7):(7)$$\varepsilon^{\prime }=2.07f^{2}-31.97f+124.64.$$

To get the permittivity of the LUT through Equation (7), the resonating frequency of the sensor loaded with the LUT can be measured by a vector network analyzer.

## 4. Experiment and Discussion

Figure 11 shows the fabricated sensor. The Agilent N5242A vector network analyzer was used for the S parameter measurement; the measurement system is shown in Figure 12.

The binary mixture of F20W/30 lubricating oil and iron powder was selected as the test object. Ten groups of 10 mL oil were placed in the measuring cylinder, with different amounts of iron powder, as shown in Figure 11. In order to reduce the influence of the environment’s temperature, the laboratory temperature was controlled at 25 °C. The dielectric constants of the lubricating oil samples with different iron powder contents are shown in Table 1 [34]

A plastic film with a thickness of 0.008 mm was added to the inner wall of the container. Ten groups of 1 mL LUT was placed in the container respectively for S_11_ parameter measuring. Each group of samples was measured for 15 times. After removing the abnormal data, the mean value was calculated. Figure 13 shows the comparison of the measured and simulated results.

Figure 13a–f demonstrates that the measuring results are basically consistent with the simulated ones. The distribution of the resonance frequency of the ten groups of samples tested combined with the fitting curve is displayed in Figure 14. It shows that the sample points are basically located on the fitting curve. According to the resonant frequency acquired by the measurement, the comparison of the permittivity calculated by Equation (7) and the reference value from Table 1 is shown in Table 2.

Referring to Table 2, using the proposed sensor to measure the dielectric constant of LUT can control the error rate within 1.5%. According to the definition of sensitivity [21,35],
(8)$$s=\frac{\Delta f[\%]}{\Delta \varepsilon_{r}}=\frac{f_{2}-f_{1}}{f_{1}(\varepsilon_{2}-\varepsilon_{1})},$$
where *f*_2_ and *f*_1_ represent the measured frequency and the initial frequency, respectively. The initial dielectric constant and measured one are *ε*_2_ and *ε*_1_. According to the measurement results, the sensitivity of the sensor is up to 3.45%. Compared to the references in Table 3, the proposed sensor has a significant advantage in sensitivity.

It is noteworthy that the dielectric constant measured is a real part. It is the limitation of the proposed sensor. This paper only focuses on the frequency of the S parameter, not on the phase and amplitude. It is found in other literatures that the complex dielectric constant can be calculated by using the amplitude and phase information of the S parameter. In the future, measuring the complex dielectric constant by the frequency, amplitude and phase of S_11_ and S_21_ will be more meaningful. In addition, the portability of the sensor is also affected by the vector network analyzer. The challenge for sensors is to develop a supporting data display and data processing modules that meet the needs of miniaturization, thereby satisfying the requirement of portable measurement.

## 5. Conclusions

In this paper, a liquid dielectric constant microwave sensor based on a cubic container structure is designed for the measurement of the liquid dielectric constant by using the resonance characteristics of CSRR and SRR. The electric field in the plane structure is spread on the side wall of the cube container structure by loading the microstrip line and SRR on the dielectric substrate. The sensitivity of the variation in the dielectric constant is enhanced. Compared with the existing work, the sensitivity of the dielectric constant measurement has been greatly improved. Compared with other market devices, the proposed sensor can achieve a high accuracy with lower costs.

## Figures and Tables

**Figure 1 sensors-20-05598-f001:**
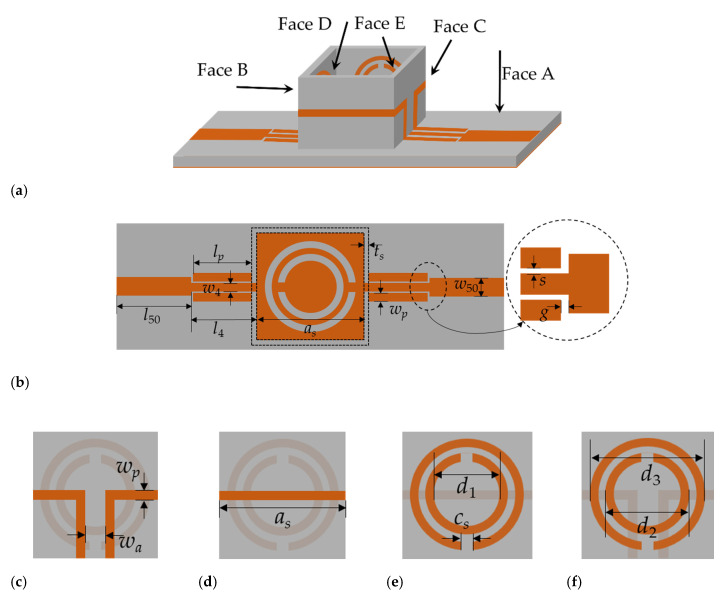
Sensor structure. (**a**) 3-D view; (**b**) Face A; (**c**) Face B; (**d**) Face C; (**e**) Face D; (**f**) Face E. The dimensions are as follows: *l*_50_ = 14.0 mm, *w*_50_ = 2.4 mm, *l_p_* = 9.6 mm, *w_p_* = 0.8 mm, *l*_4_ = 10.0 mm, *w*_4_ = 1.0mm, *a_s_* = 12.3 mm, *t_s_* = 0.127 mm, *s* = 0.1 mm, *g* = 0.2 mm, *h_s_* = 6.5 mm, *w_a_* = 12.6 mm, *d*_1_ = 4.0 mm, *d*_2_ = 4.4 mm, *d*_3_ = 5.2 mm, and *c_s_* = 1.1 mm.

**Figure 2 sensors-20-05598-f002:**
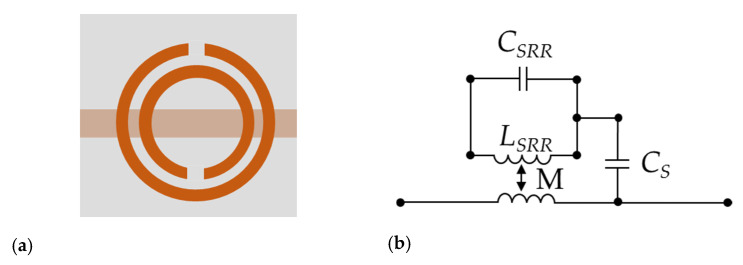
(**a**) The structure of the split resonance ring (SRR). (**b**) The equivalent circuit.

**Figure 3 sensors-20-05598-f003:**
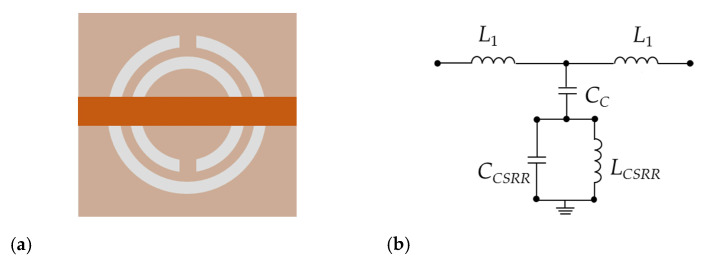
(**a**) The structure of the complementary split resonance ring (CSRR). (**b**) The equivalent circuit.

**Figure 4 sensors-20-05598-f004:**
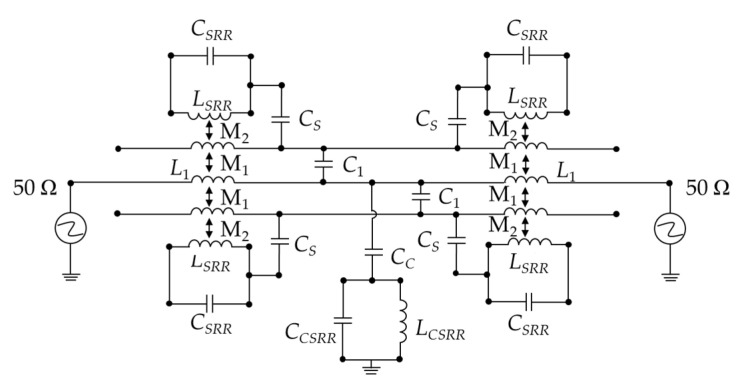
The equivalent circuit of the sensor. The electrical parameters are *C_SRR_* = 0.258 pF, *L_SRR_* = 1.031 nH, *C_S_* = 0.157 pF, *L*_1_ = 3.471nH, *C*_1_ = 0.347 nH, *Cc_SRR_* = 0.258 pF, *L_CSRR_* = 1.031 nH, *Cc* = 0.157 pF, and *C_SRR_* = 0.258 PF.

**Figure 5 sensors-20-05598-f005:**
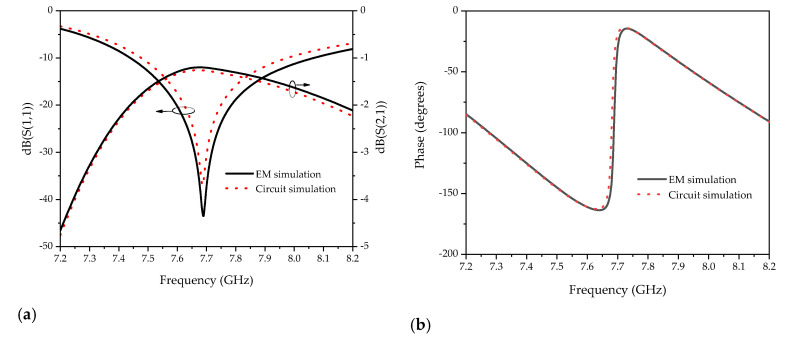
The comparisons of the electromagnetic simulation and the equivalent circuit simulation: (**a**) magnitude response; (**b**) phase response.

**Figure 6 sensors-20-05598-f006:**
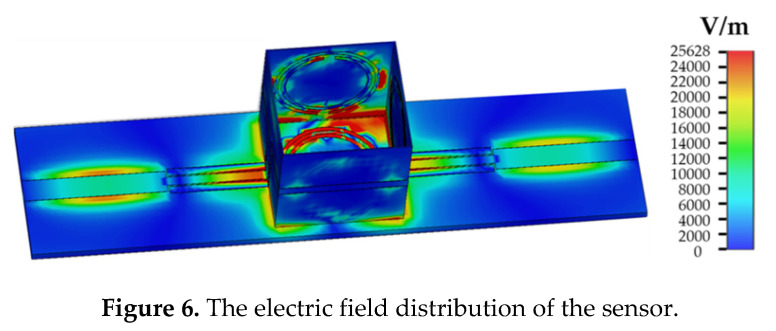
The electric field distribution of the sensor.

**Figure 7 sensors-20-05598-f007:**
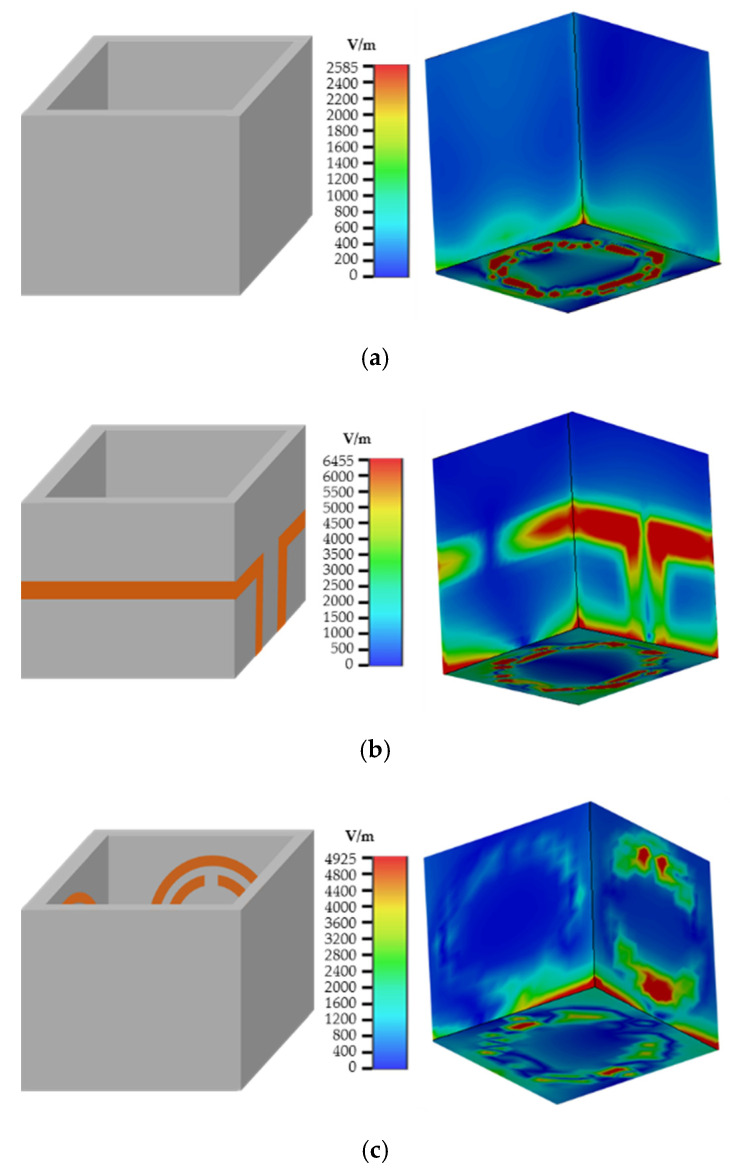
The electric field distribution: (**a**) without a microstrip line and SRR; (**b**) only microstrip lines loaded; (**c**) only SRR loaded; (**d**) microstrip line and SRR loaded.

**Figure 8 sensors-20-05598-f008:**
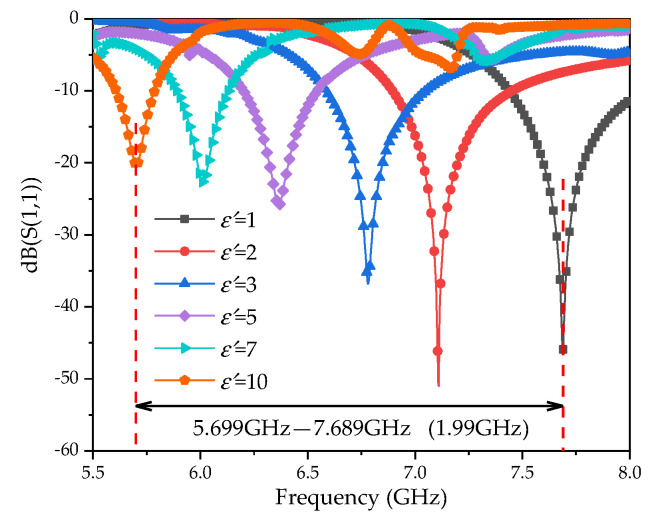
Simulated reflection coefficients under various dielectric constants of the liquid under test (LUT).

**Figure 9 sensors-20-05598-f009:**
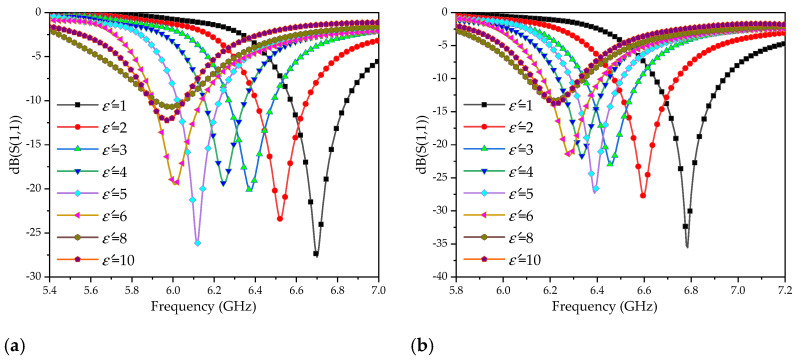
Simulated reflection coefficients under various dielectric constants of the LUT with ordinary containers: (**a**) glass container; (**b**) plastic container.

**Figure 10 sensors-20-05598-f010:**
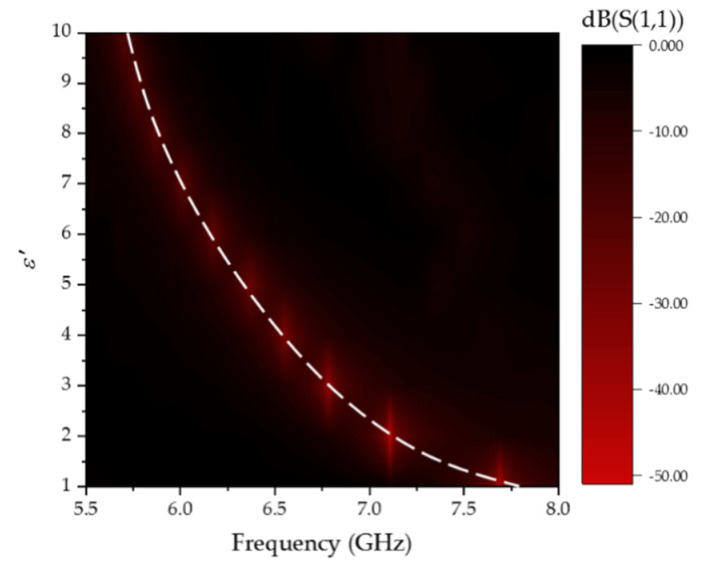
The resonant frequency curve varies with the permittivity.

**Figure 11 sensors-20-05598-f011:**
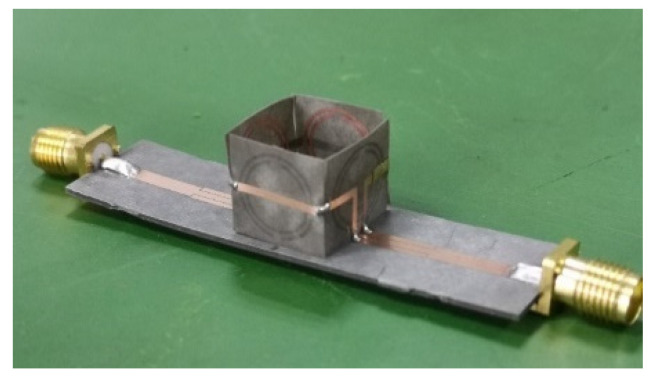
The fabricated sensor.

**Figure 12 sensors-20-05598-f012:**
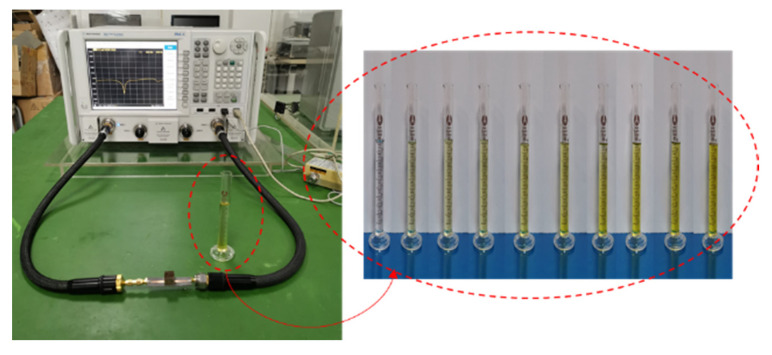
The measurement system for the fabricated sensor.

**Figure 13 sensors-20-05598-f013:**
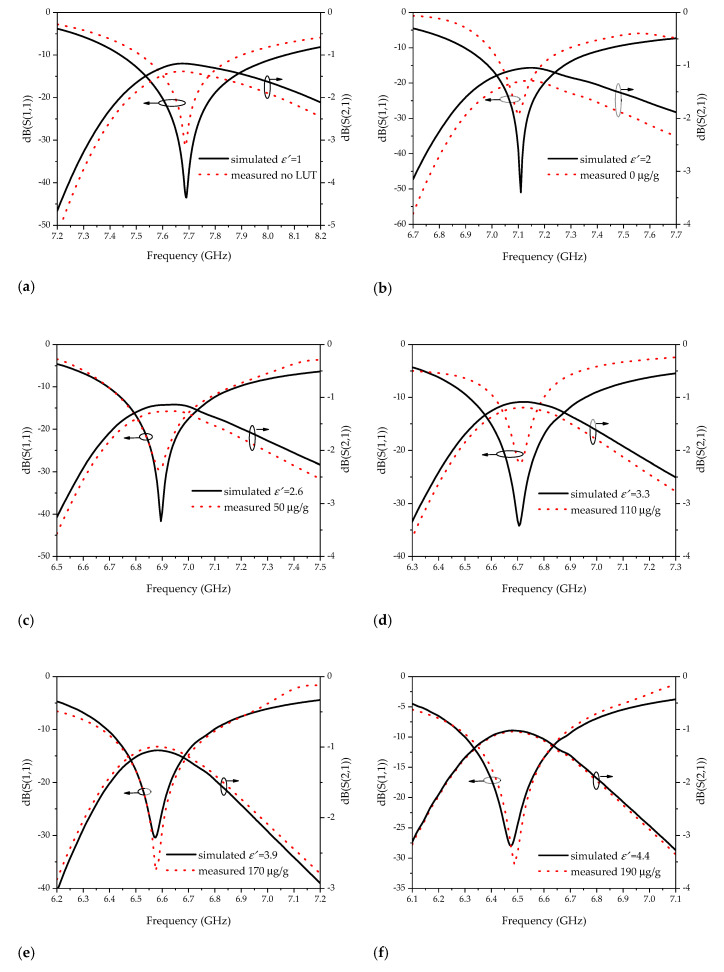
Comparison of the simulation and measured results: (**a**) *ε*′ = 1, no ULT; (**b**) *ε*′ = 2, content = 0 μg/g; (**c**) *ε*′ = 2.6, content = 50 μg/g; (**d**) *ε*′ = 3.3, content = 110 μg/g; (**e**) *ε*′ = 3.9, content = 170 μg/g; (**f**) *ε*′ = 4.4, content = 190 μg/g.

**Figure 14 sensors-20-05598-f014:**
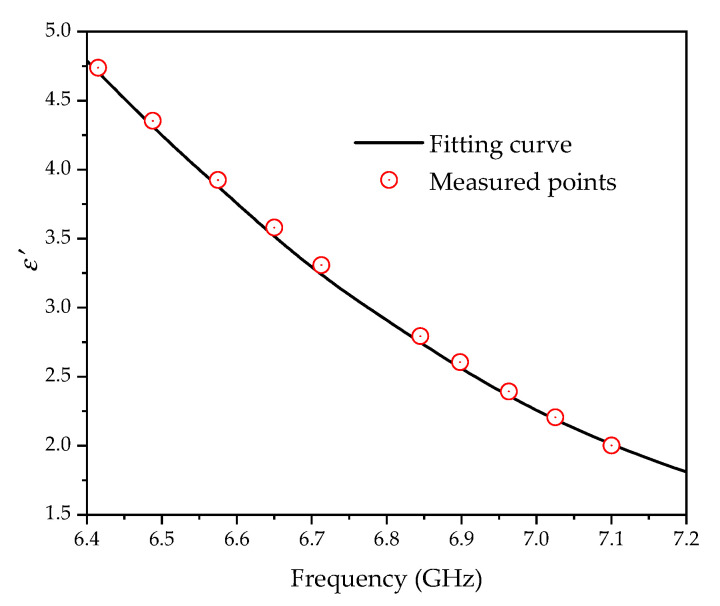
The distribution of the measured points in the fitting curve.

**Table 1 sensors-20-05598-t001:** The dielectric constants of the lubricating oil samples with different iron powder contents.

No	1	2	3	4	5	6	7	8	9	10
Content (μg/g)	0	20	40	50	70	110	130	170	190	250
Permittivity	2	2.2	2.4	2.6	2.8	3.3	3.6	3.9	4.4	4.8

**Table 2 sensors-20-05598-t002:** Comparison of the measurement results.

No	1	2	3	4	5	6	7	8	9	10
*ε*′ (Reference)	2	2.2	2.4	2.6	2.8	3.3	3.6	3.9	4.4	4.8
*ε*′ (Measured)	2.00	2.21	2.39	2.61	2.79	3.31	3.58	3.92	4.35	4.74
Relative error %	0.09	0.30	0.28	0.25	0.24	0.26	0.55	0.63	1.06	1.30

**Table 3 sensors-20-05598-t003:** Comparison of sensitivity.

Approach	Type	Q-Factor	Sensitivity
[18]	submersible	252	2.20%
[19]	submersible	59	3.04%
[20]	container	42	0.27%
[21]	container	25	1.00%
Proposed Sensor	container	73.3	3.45%

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
