# Peer review of "Design of a High Sensitivity Microwave Sensor for Liquid Dielectric Constant Measurement"

_sensors, 2020, doi:10.3390/s20195598_

Round 1
Reviewer 1 Report
The paper presents a non-planar SRR for liquid detection. The idea of using SRR for detection of liquid is well explored and the only merit to this paper is that the SRR is not planar. The paper can be accepted after addressing the following questions/concerns.
1) The introduction lacks adequate literature, I personally suggest including a table comparing the sensitivity, Q-factor etc of the presented sensor with other SRR based (planar or non-planar) liquid sensors presented in the literature.
2) The authors stated "SRR is a kind of small electronic resonator. It is a metamaterial structure with both negative permeability and negative dielectric constant." Please elaborate.
3) Equation 2 is a generic SRR equation that does not take into account fringing capacitance due to a ground plane or metal at the back (For example Capacitance between rings in Face D and the microstrip in Face B). Without these details, it does not completely represent the presented structure. Please re-work the theory to include all the effects of fringing capacitances. Also for the CSRR, similar re-work is required. If the actual circuit is not represented, I fail to understand the need for section 2 if it is just taken from literature.
3) Figure 5 is simulated at which frequency? Figures 5, 6, 7, & 8, please increase the font size of legends.
4) Add a figure that shows a comparison between unloaded (No LUT) sensors simulated and measured S11. Since it is a two-port measurement also include S21. Calculate the Q-factor and mention it in the manuscript. A lot of information will be available in phase information, so I personally suggest plotting the phase as well for the measured data along with magnitude.
5) Comment on the repeatability of the measurements.
6) Add a section that includes the challenges and limitations of the proposed sensor.
Other comments:
- Please keep the font size similar in all the figures.
- There are a number of grammatical errors, please correct them before re-submission.
Reviewer 2 Report
This paper presented a method to utilize microwave resonator to detect liquid dielectric constant. This paper is organized well with sufficient theory, simulation as well as experimental results. Please see my comments as below,
1). Need more clear description on how structure on face B, C, D, E and F impact the resonate frequency and why need structures on such faces?
2). Many published papers already described different type of resonators, Can authors provide the advantage and disadvantages of this structure?
Reviewer 3 Report
The manuscript “Design of the high sensitivity microwave sensor for liquid dielectric constant measurement” numerically and experimentally demonstrates a container-based sensor to characterize the dielectric characteristic of the binary mixture of F20W/30 lubricating oil and iron powder and obtain a relatively high sensitivity of 3.45%.There is undeniably some novel work on the container-based sensor. The experimental research and verification work by author can help readers understand the dielectric characterization technology to some extent. However, the writing logic, organization and explanation of the article are poor. Moreover, the article is poorly readable in English, there are many grammar problems. The readability should be improved by a native English speaker. The author is asked to give detailed theoretical analysis and explanation and are encouraged to highlight the main contributions of their work. My Major comments are listed below: 1) The introduction part is too loose and too messy to provide good help to potential readers, and must be improved. There is no objective and comprehensive description of the research status of microwave-technology-based dielectric constant measurement, including the pros and cons of various microwave measurement technologies. The background description of the test object is not explained. It is recommended to rewrite the introduction. 2) In the “Design and Analysis” section, line 56 of the paper directly gives the conclusion “three-dimensional distribution of the electric field is more sensitive to the variation of dielectric constant of LUT”. The conclusion is either drawn through experiments and theoretical analysis or from the conclusion of published papers (references should be marked). The same problem, such as "It dramatically enhances the electric field intensity in the measurement area and improves the sensitivity of the sensors" in line 69. What about the evidence? 3) The theoretical part of sensing, such as perturbation, should be added. Essentially, the improvement of the sensitivity of the proposed structure is due to the fact that loading SRR the side wall of the container can increase the effective volume ratio of perturbation. The sensing theory explanation and understanding should be strengthened. 4) Lines 76-95 only schematically show the equivalent circuit models of SRR and CSRR resonators, and have not effectively contribution to the analysis of the experimental results. The author should give the equivalent LC circuit model of the container-based sensor, loading CSRR on the bottom substrate and SRR on the side wall of the container, and give the comparison of the electromagnetic simulation and the ADS model, in order to help readers better understand the working mechanism of the sensor. 5) The description of the device structure is unclear, such as line 65, 0.008mm plastic film, what’s the purpose? I am also confused by the description "fed by two parallel coupling lines at the λ/4 step impedance (SIR)" in line 67. Additionally, why did the author analyze the resonance characteristics of λ/4 shorted SIR, and the shorted one isn’t used in the sensor. In my opinion, SIR only plays the role of impedance matching, therefore the formula analysis of SIR does not contribute to the sensing performances and suggest to be deleted. 6) Detailed assembly instructions of the container-based sensor should be given by use of PCB board. The corresponding explanation is too simple and easily confusing. 7) Line 192, there is a spelling error: "Significant" should be "significant".Author Response
Please see the attachment.

Reviewer 4 Report
In my opinion the manuscript is interesting but the authors should underline the following items: 1) the use of SRR as metamaterials to affect the electric field is well know, the manuscript novelty should be underlined; 2) the comparison with other technique should be given even with with refernce to available on market large frequency/ purpose devices (probe by hp or DAK probes)
Round 2
Reviewer 3 Report
1) The description of Figure 2(a) and Figure 3(a) is incorrect. It should be a transmission line loaded with SRR and CSRR instead of SRR and CSRR, respectively. Because the Figure 2(b) and Figure 3(b) is the equivalent circuit model of a transmission line loaded with CSRR and a CSRR, respectively. The author is too rigorous. 2) The author must give the specific parameter values of lumped circuit equivalent model of the sensor in Figure.4. 3) I have another question. Generally, the transmission line structure loaded with CSRR will produce zero in S21, why the structure given by the author in S11? Please give a detailed explanation. 4) There are still many grammatical mistakes. For example: line 94, "The film is easy to be replaced" is very confusing, I don’t know what the author wants to express. Line 105-109, the description of the negative permittivity and permeability related to SRR has little contribution to this article, and should be deleted. 5) Lines 141-143, Equation 6 is the Q value formula for the single-port device. But the paper sensor structure is a dual-port device, why the Equation 6 is used by author to calculate the Q value. 6) Line 145, it should be According to Equation.6 instead of Equation 2. 7) Reference 11 uses the resonance method, not the transmission method at all. The standard transmission method uses the transmission characteristics of S21 (the phase and amplitude) to characterize the dielectric properties of materials. Authors should cite reference objectively.Author Response
Please see the attachment.
